# Global impacts of future cropland expansion and intensification on agricultural markets and biodiversity

Florian Zabel [1,8], Ruth Delzeit [2,8], Julia M. Schneider [1], Ralf Seppelt [3,4,5], Wolfram Mauser[1] & Tomáš Václavík [3,6,7,8]

With rising demand for biomass, cropland expansion and intensification represent the main strategies to boost agricultural production, but are also major drivers of biodiversity decline. We investigate the consequences of attaining equal global production gains by 2030, either by cropland expansion or intensification, and analyse their impacts on agricultural markets and biodiversity. We find that both scenarios lead to lower crop prices across the world, even in regions where production decreases. Cropland expansion mostly affects biodiversity hotspots in Central and South America, while cropland intensification threatens biodiversity especially in Sub-Saharan Africa, India and China. Our results suggest that production gains will occur at the costs of biodiversity predominantly in developing tropical regions, while Europe and North America benefit from lower world market prices without putting their own biodiversity at risk. By identifying hotspots of potential future conflicts, we demonstrate where conservation prioritization is needed to balance agricultural production with conservation goals.

[1] Department of Geography, Ludwig-Maximilians-Universität München, 80333 Munich, Germany. [2] Kiel Institute for the World Economy, 24105 Kiel, Germany. [3] Department of Computational Landscape Ecology, UFZ—Helmholtz Centre for Environmental Research, 04318 Leipzig, Germany. [4] Institute of Geoscience & Geography, Martin-Luther-University Halle-Wittenberg, 06099 Halle (Saale), Germany. [5] iDiv—German Centre for Integrative Biodiversity Research, 04103 Leipzig, Germany. [6] Faculty of Science, Department of Ecology and Environmental Sciences, Palacký University Olomouc, 78371 Olomouc, Czech Republic. [7] Global Change Research Institute of the Czech Academy of Sciences (CzechGlobe), 60300 Brno, Czech Republic. [8]These authors contributed equally: Florian Zabel, Ruth Delzeit, Tomáš Václavík. Correspondence and requests for materials should be addressed to R.D. (email: ruth.delzeit@ifw-kiel.de)

For thousands of years, humans have cultivated the planet to satisfy their needs for food, fibre and energy. Today, farmlands dominate 38% of the global land surface[1] and almost 30% of global net primary production is appropriated for human use[2]. The pace of farmland production growth is unlikely to continue[3], but the demand for agricultural commodities is projected to increase inexorably (70–100% by 2050)[4,5]. Since the focus on agricultural production is motivated also by income generation and economic growth, high pressure on farming systems will continue in the next decades[6–8].

As a result, agriculture is likely to remain the primary driver of global biodiversity loss, because both strategies to increase production, namely cropland expansion and intensification, pose major threats to many common as well as IUCN red-listed species[9,10]. While cropland expansion into uncultivated areas threatens biodiversity mainly through the loss and fragmentation of natural habitat[11,12], the negative effects of cropland intensification may be less pronounced[13]. There is clear evidence, however, that land-use intensification threatens multiple taxa of primarily farmland species due to habitat homogenisation[14,15], irrigation[16] and high inputs of agro-chemicals[17,18], such as fertilisers and pesticides. Therefore, meeting future biomass demands while, at the same time, safeguarding remaining ecosystems and biodiversity is a critical challenge we face in the 21st century[19] (Sustainable Development Goals 2, 12 and 15[20]).

Recent advances in data availability[21–23] and spatially explicit modelling of land systems[24–26] improved our ability to assess future agricultural impacts. General solutions to cope with the increasing demand for agricultural resources have been proposed[27,28] but the spatial relationship between different farming strategies and biodiversity patterns have been understudied. Although cropland expansion and intensification often occur simultaneously, recent studies evaluated only one aspect separately or did not separate intensification from expansion processes[29–33]. Often a limited set of production metrics was used (e.g. yields[34,35]) or biophysical constraints of farmland production were considered but socio–economic drivers were ignored or vice versa[36,37]. Changes in agricultural productivity are addressed in some scenario studies feeding yield changes into partial or general equilibrium models[38–40], but feedbacks from the economic model to biophysical models are neglected. Thus, emerging trade-offs have not yet been addressed using comparable scenarios that integrate biophysical and socio–economic drivers of crop production[12,34,41–44]. Therefore, integrated approaches are required that

(i) utilise comparable scenarios of both cropland expansion and intensification,
(ii) account for spatial information on biophysical constraints as well as socio–economic drivers of agricultural production,
(iii) capture repercussions of changes in supply and demand on regional and global markets, and
(iv) estimate how different farming strategies and their impacts on biodiversity play out across space.

This is crucial to assess the feasibility of achieving desired agricultural pathways and minimise their impact on areas with the highest conservation value.

Here we capture feedbacks between biophysical and socio–economic drivers of land-use change as well as interactions with biodiversity. We examine global trade-offs between agricultural markets and global biodiversity that future farmland production may impose (Fig. 1). First, we combine two established approaches from previous work of the authors[6,29], which integrate both biophysical and socio–economic conditions to create maps of future cropland expansion and intensification

potentials simulated for 17 major agricultural crops at 30 arc-sec spatial resolution (see Supplementary Notes 1, 2, 3). These crops represent 73% of global cropland area and crop production[45] and cover the most important staple and energy crops, to also capture trends in political support of biofuels.

These integrated potentials of cropland expansion and intensification account for the interplay of biophysical constraints at the local scale, such as water availability, soil quality and climate change, and regional socio–economic drivers, such as population growth and dynamics in consumption patterns. Second, we examine the impact of cropland expansion and intensification on agricultural markets (Supplementary Note 4). To do so, we apply a computable general equilibrium (CGE) model of the world economy that accounts for interlinkages between economic sectors to two comparable scenarios of cropland expansion and intensification until 2030. These are compared to a reference scenario that carries forward current trends in population growth, gross domestic product and trade policies[46]. The cropland expansion scenario allows additional 7.3 million km² of land to be available for crop production in areas with the highest 10% of global expansion potential. Comparably, the cropland intensification scenario allows closing yield gaps on 10% of land with the highest global intensification potential, up to the level that both scenarios leads to equivalent global production gains (Supplementary Note 1). Finally, we use global range maps for 19,978 vertebrate species to examine the spatial concordance between patterns of global biodiversity and potentials for near-future cropland expansion and intensification (see Supplementary Notes 5, 6). Our goal is to (1) quantify the relative differences in the impact of alternative global farming strategies (cropland expansion vs. intensification) on crop yields, prices, trade and consumption, and to (2) identify hotspots of potential future conflicts between cropland expansion, intensification and biodiversity.

## Results

**Impacts on agricultural markets.** Both farming strategies resulted in additional 19% of global crop production compared to the reference scenario. While in the expansion scenario, an area of 7.3 million km² is additionally used for crop production, an area of 1.5 million km² is intensified in the intensification scenario. Both strategies had different impacts on considered geographical regions. When compared to the reference scenario in 2030, the changes in production under the cropland expansion mirrored the relative changes in cropland expansion area (Supplementary Fig. 4): crop production increased most in South and Central America (+146%) and in Australia/New Zealand (+78%) (Fig. 2). Due to the increase in supply in agricultural markets, crop prices fell in all regions, including those regions where domestic production decreased (e.g. EU, USA, Russia). The EU turned from a net-exporter to a net-importer, while the net-importer Russia increased imports due to lower world market prices (Supplementary Fig. 15).

Cropland intensification caused the strongest increases in production in Sub-Saharan Africa (+78%), India (+68%), and Former Soviet Union (+63%) (Fig. 2). Crop prices dropped strongest in regions with high total intensification potentials and high shares of top 10% areas on total land endowment such as Sub-Saharan Africa and India (Supplementary Fig. 8), while others (e.g. EU and Middle East and Northern Africa) also benefited from lower world market prices.

The comparison of the expansion and intensification scenarios showed an increase in crop production, e.g. in Sub-Saharan Africa and Australia, but it also showed contradicting impacts on several regions: crop production increased significantly in Central and

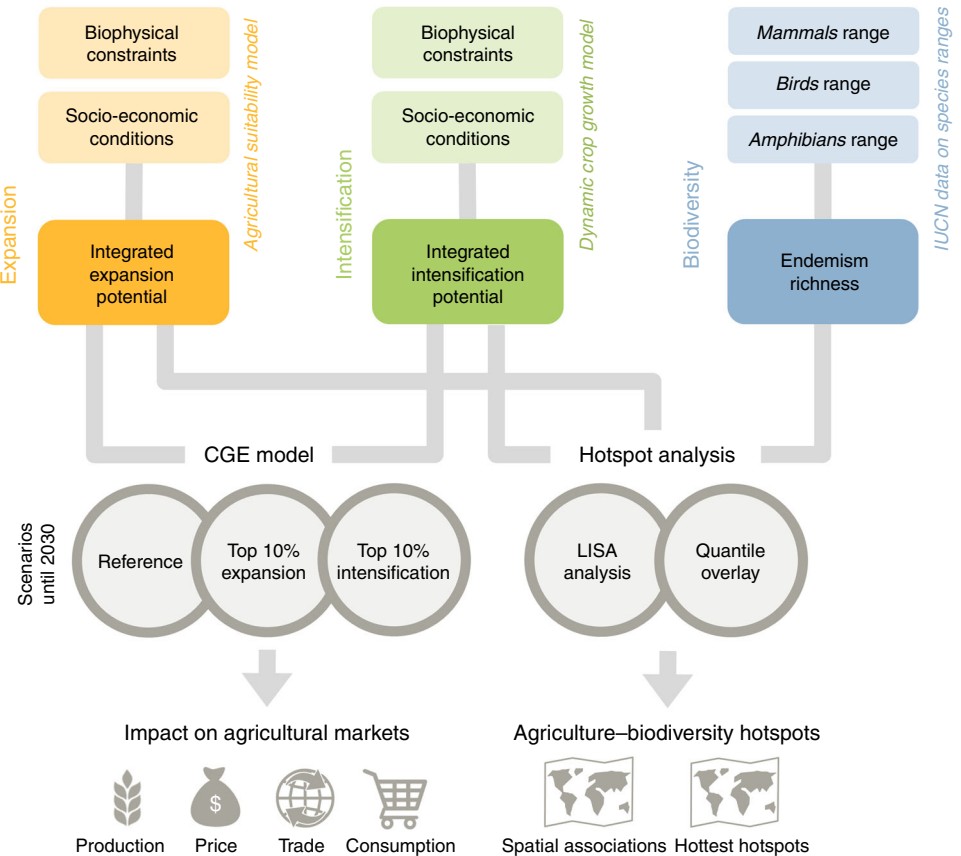

**Fig. 1** Overview of the study design. The study is based on three sources of data on global cropland expansion, intensification and biodiversity. Both maps of cropland expansion potential and intensification potential are simulated for 17 major agricultural crops at 30 arc sec resolution and integrate information on biophysical constraints (e.g. topography, soil quality, climate change) and socio–economic conditions (e.g. population growth, consumption patterns). The integrated cropland expansion potential is developed by a model of near-future agricultural suitability, while the integrated cropland intensification potential is developed by a dynamic crop growth model. A computable general equilibrium (CGE) model of the world economy, applied to two scenarios of cropland expansion and intensification until 2030, quantifies the impact on agricultural markets in terms of crop production, price, trade and consumption. We use a reference scenario up to 2030 for reference that carries forward current trends in population growth, gross domestic product and trade policies. Endemism richness integrates IUCN range maps of 19,978 species of mammals, birds and amphibians into a global biodiversity metric aggregated at 55-km resolution of an equal-area grid. This metric combines species richness with a measure of endemism (i.e. the range sizes of species within an assemblage) and thus indicates the relative importance of a site for conservation. Hotspot analysis, using Local Indicator of Spatial Association (LISA) and quantile overlay, identifies hotspots where global biodiversity is most affected by near-future cropland expansion and intensification

South American countries under the cropland expansion scenario, while crop production decreased under the intensification scenario. The opposite effect appeared, e.g. in India and China. While the intensification scenario caused crop production in these regions to increase by 68 and 5%, respectively, the low expansion potentials caused crop production to decrease by 2 and 3% under the expansion scenario. Notably, India was a net-importer of crops under the expansion scenario, while it was a net-exporter under the intensification scenario (Supplementary Fig. 15).

One would expect that given relatively large cropland intensification potentials (Supplementary Fig. 8), Sub-Saharan Africa would increase crop consumption more than, e.g. China with lower cropland intensification potentials. However, with a stronger economic growth compared to Sub-Saharan Africa, China increased its net imports of crops such that food consumption increased stronger than domestic production (Supplementary Figs. 15, 16). Hence, the impacts of farming strategies on agricultural markets did not only depend on the changes in land productivity or land endowment, but were a result from market mechanisms, since the economies compete under flexible prices on global markets.

**Agriculture–biodiversity hotspots**. Both farming strategies resulted in equal global production gains of 19% more crop yields than the reference scenario (Fig. 2) but differed substantially in their impact on biodiversity. To understand how cropland expansion, intensification and biodiversity are interlinked, we first examined statistically significant spatial associations between gradients of estimated agricultural potentials in 2030 and endemism richness for expansion and intensification separately (Fig. 3).

The hotspot regions where high biodiversity will be most threatened by cropland expansion or intensification in 2030 were found overwhelmingly in the tropics, with cropland expansion affecting larger areas than cropland intensification (significant hotspots covering 14 and 8% of the terrestrial ecosystems, respectively). Biodiversity hotspots under cropland expansion pressure occurred in Central and South America, including the western part of the Amazon Basin and the Atlantic forest, in the forests and savannahs of Central Africa and Madagascar, as well as in parts of South Africa, Eastern Australia and large portions of South-East Asia (Fig. 3a). The cropland intensification pressure on biodiversity was generally less pronounced, especially in Latin America, but included regions in Sub-Saharan Africa, India,

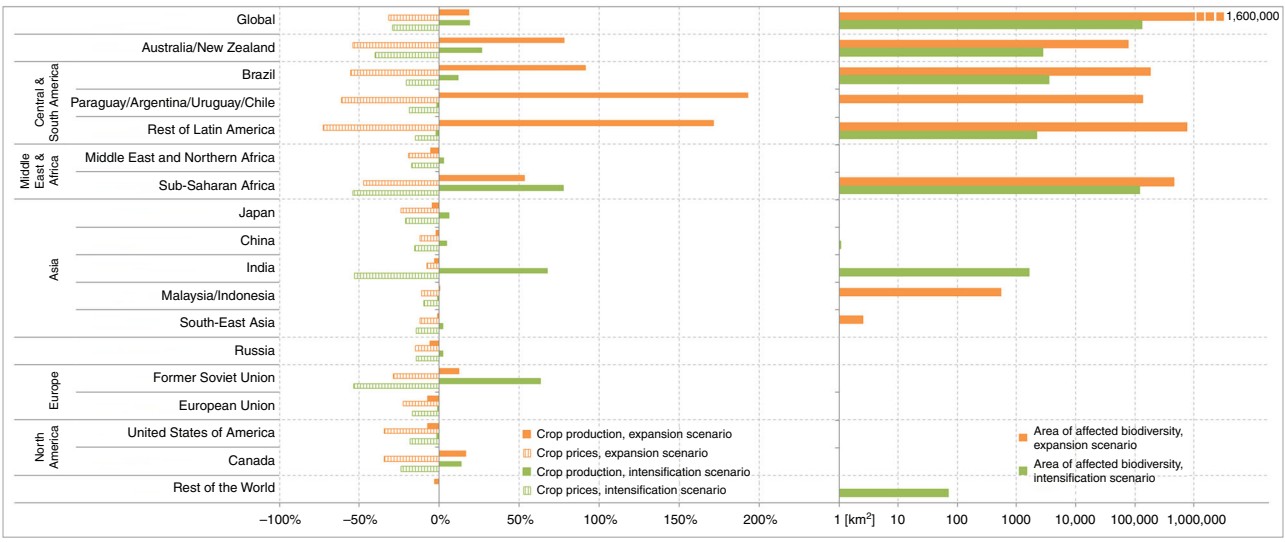

**Fig. 2** Impacts of expansion and intensification on production, prices and area of affected biodiversity. The left panel shows the change in crop production and prices under expansion and intensification scenarios compared to the reference scenario in 2030, accounting for current trends in population growth, gross domestic product and trade policies. The right panel shows the area of land where the top 10% of the most biodiverse regions are threatened under the expansion and intensification scenario (x-axis for area is scaled logarithmically)

Nepal, Myanmar and China where farming intensity was projected to increase substantially until 2030 (Fig. 3b). While hotspot patterns for birds and mammals showed high spatial agreement (64 and 66% overlap for cropland expansion and intensification, respectively), the areas of high agricultural potentials associated with high endemism richness were relatively smaller for amphibians (41 and 40% overlap with the other taxa) due to their smaller ranges concentrated in specific geographical areas.

Agricultural intensification affects species not only in croplands but also in surrounding habitats, thus the impact will likely differ for species with different habitat requirements (e.g. forest specialists are unlikely to reside in significant numbers within existing farmland). Consequently, we used information on species preferred habitat types and examined spatial associations between intensification and biodiversity for (1) species that are regular or at least marginal cropland users vs. (2) forest or natural habitat specialists (Fig. 4). As expected, the intensification pressure was more pronounced for cropland users (significant hotspots covering 8% of the terrestrial ecosystems; Fig. 4a) than for forest or natural habitat specialists (4%, Fig. 4b), especially in the Chaco ecoregion of South America, Central and Eastern Africa and Southern Asia. However, the general hotspot patterns remained largely consistent, suggesting that areas with high endemism richness in general hold high diversity of forest or natural habitat specialists as well as high diversity of cropland users.

The associations of low agricultural potentials and low endemism richness (i.e. cold spots) showed consistent patterns for both scenarios across all three taxonomic groups (Fig. 3). The cold spots were identified mostly on non-arable, desert, or ice-covered land, but also in industrialised agriculture in North America and Western Europe, where further increases of yields are limited (53 and 48% of land surface for expansion and intensification, respectively). Other regions with high agricultural potentials were either not significant or occurred in areas with comparatively low biodiversity (high–low associations), e.g. the Midwest of North America, Former Soviet Union, Sub-Saharan Africa (Fig. 3a; 9% of terrestrial ecosystems) and, for the intensification scenario, also large parts of India and China (Fig. 3b; 15% of terrestrial ecosystems), where our simulations

show high production gains in the intensification scenario (Fig. 2). However, these high intensification regions with relatively low global biodiversity were much smaller when focusing on cropland species (5% of land surface) as opposed to forest or natural habitat specialists (11% of land surface), being restricted mostly to Former Soviet Union and China (Fig. 4a).

We then examined the same top 10% areas for cropland expansion and intensification as in the economic analysis and overlaid them with the biodiversity data above the 10th percentile to identify regions where the highest endemism richness coincides with the highest potential pressure from land expansion and intensification.

These 'hottest hotspots' where the highest biodiversity may be particularly threatened by future cropland expansion and intensification, were found especially in Central and South America (affecting an area of 1.097 million km²), Sub-Saharan Africa (773,375 km²) and Australia (79,490 km²). Cropland expansion affected biodiversity hotspots mainly along the tropical Andes, the Brazilian Atlantic forest and in West and East Africa (Fig. 5). For the intensification scenario, the areas with the highest risk of biodiversity loss were located in Sub-Saharan Africa (122,702 km²) and Brazil (3,560 km²). In total, cropland expansion was likely to affect much larger areas (1.6 million km²) with the highest conservation value than cropland intensification (132,984 km²) (Figs. 2 and 5b, Supplementary Table 5). For intensification, these potential conflict areas, however, would be 2.4 times larger for cropland species than for forest and natural habitat specialists (Supplementary Fig. 19). On the other hand, large areas with high cropland expansion potential exist in Paraguay, Argentina, Uruguay and Chile (1.176 million km²), Sub-Saharan Africa (894,178 km²) and Brazil (871,759 km²), that do not overlap with the top 10% values of biodiversity (Supplementary Table 5). Sub-Saharan Africa also holds most of the top 10% areas for intensification (673,300 km²) that do not at the same time belong to the top 10% of biodiverse areas.

Changing the (arbitrary) percentile threshold from 5 to 30% led to a considerable increase in the conflict area (especially in Latin America for expansion and in East Africa and South Asia for intensification; Supplementary Fig. 18), with a substantially steeper slope for expansion than for intensification (Fig. 5b).

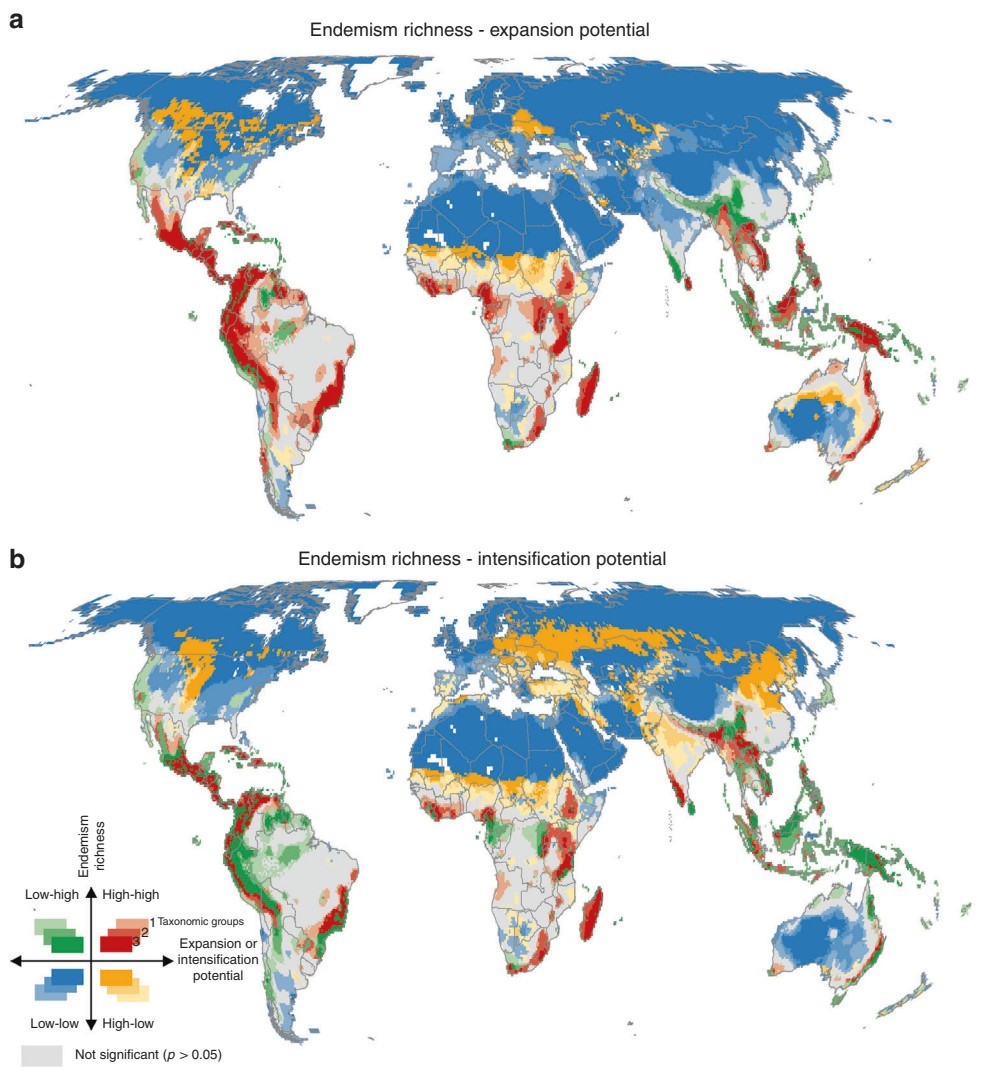

**Fig. 3** Spatial association between endemism richness and potentials for **a** cropland expansion and **b** intensification. They are calculated using local indicators of spatial association (LISA) at 55-km resolution. High–high clusters indicate hotspot locations (red), in which areas most suitable for expansion/intensification of cropland are significantly associated with high values of endemism richness (at 0.05 significance level). Low–low clusters (blue) show cold spot locations, in which areas with low potential for expansion/intensification are associated with low values of endemism richness. High–low and low–high clusters show inverse spatial association. Three shades of colours indicate significant results for one, two or all three taxonomic groups (birds, mammals, amphibians)

## Discussion

Here, we applied an iterative coupling approach[6], accounting for both cropland expansion and intensification specifically designed to be equivalent in terms of reaching the same production targets. For consistency, we assumed neither costs of expanding cropland nor costs for intensifying production. We quantified the impact of both strategies on agricultural markets by taking trade as well as feedbacks between supply and demand into account and identified areas most susceptible to biodiversity loss, using an integrated approach that combined global economic analysis with fine-scale agro-ecological model simulations (30 arc-sec resolution) and a broader-scale biodiversity measure (55-km resolution).

Our analyses showed substantial trade-offs between cropland expansion and cropland intensification scenarios and their impacts on crop production and biodiversity. From an economic point of view, both scenarios contributed to improved food security in terms of increased production and lower prices not only in places where crop production rose (e.g. Sub-Saharan Africa or Australia under both scenarios) but also in regions that increased import of crops due to lower world market prices. However, contradicting impacts were apparent in several regions, most notably in Latin America with rising production under the cropland expansion scenario and decreasing production under the intensification scenario, or in India and China with the opposite effects. In addition, we saw contrasting impacts on trade flows under the two scenarios: The European Union and India turned from a net-importer in the cropland expansion scenario to a net-exporter in the intensification scenario. With respect to food consumption, regions affected by food insecurities (e.g. South-East Asia and Sub-Saharan Africa) changed consumption to a different degree under the two scenarios. Food consumption in India and to a smaller degree in Sub-Saharan Africa increased more under the intensification scenario (+35 and 28%, respectively) compared to the expansion scenario (+4 and 21%, respectively), while in the rest of South-East Asia both scenarios resulted in an increase of about 7% (Supplementary Fig. 16). For South and Central American countries, the cropland expansion scenario is clearly the beneficial strategy with respect to food security.

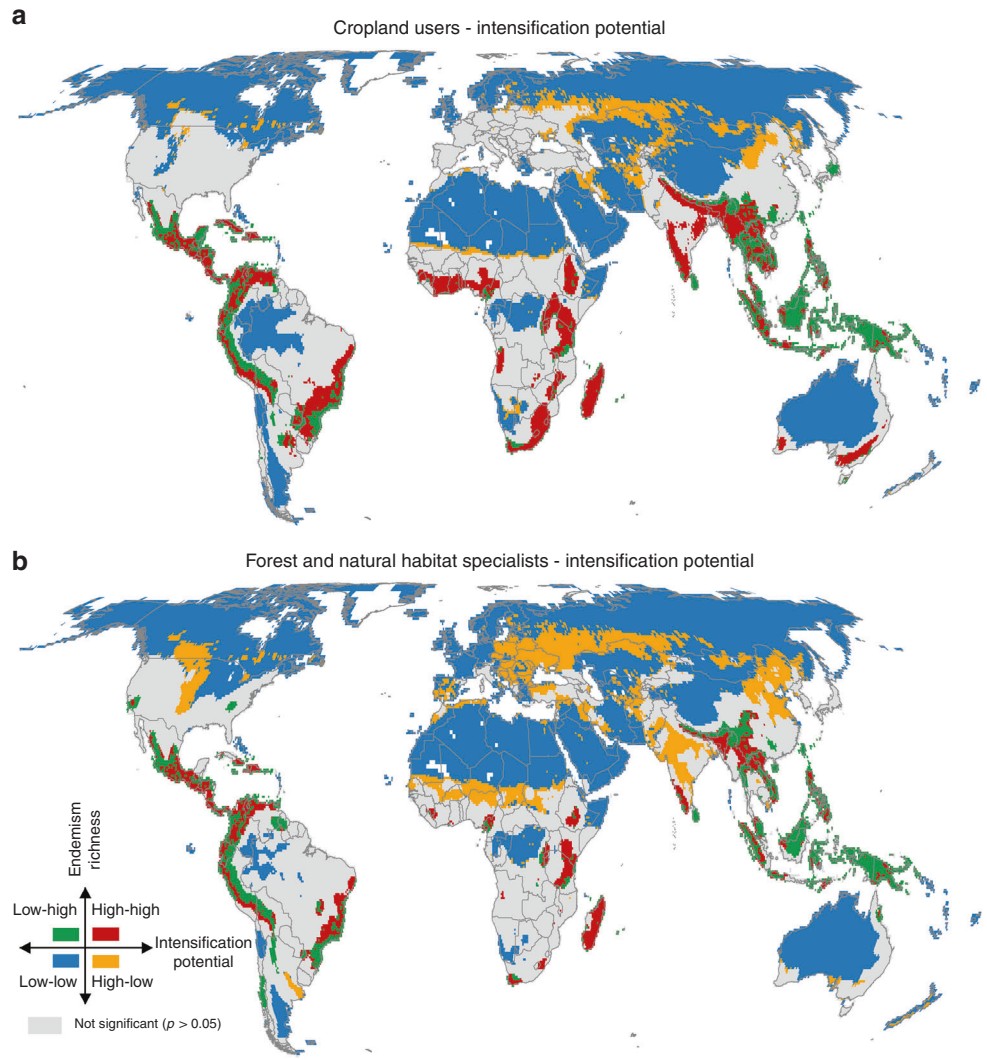

**Fig. 4** Spatial association between potentials for cropland intensification and endemism richness for **a** regular or marginal cropland users and **b** forest or natural habitat specialists. They are calculated using local indicators of spatial association (LISA) at 55-km resolution. High–high clusters indicate hotspot locations (red), in which areas most suitable for intensification of cropland are significantly associated with high values of cropland users/forest or natural habitat specialists (at 0.05 significance level). Low–low clusters (blue) show cold spot locations, in which areas with low potential for intensification are associated with low values of cropland users/forest or natural habitat specialists. High–low and low–high clusters show inverse spatial association

From a biodiversity point of view, the projected cropland expansion and intensification will likely occur in many regions that are valuable for biodiversity conservation. These pressure hotspots were found mostly in tropical ecosystems of Latin America, Central Africa and South-East Asia that were previously identified as areas where biodiversity is most threatened by agricultural production[29,31,35,47]. However, our calculations highlighted different hotspots of potential future conflict for the two agricultural pathways. Cropland expansion threatened biodiversity most in Latin America and Central Africa that contain large, relatively intact natural habitats with biophysical and socio–economic conditions likely to promote cropland expansion in the next decades. On the other hand, cropland intensification is likely to affect considerably smaller areas with the highest endemism richness in comparison to cropland expansion (~20-fold difference for the 10% threshold). But these top-pressure places include often overlooked regions in India, Myanmar or East Africa where existing small-scale cropping systems have high potential for further intensification but in the same time harbour substantial biodiversity, typically under no form of formal protection.

Indeed, we found relatively little coverage of our agriculture–biodiversity hotspots by terrestrial protected areas listed in the World Database on Protected Areas (WDPA, IUCN categories I-VI). The overlap of the hottest hotspots with the WDPA[48] showed that <35% (625,000 km²) of these hotspots are currently protected. While almost half of these areas are under strict protection (agriculture restricted; IUCN class Ia, Ib, II), the other half is less strictly protected (agriculture partly allowed; IUCN classes III, IV, V, VI). However, more than 65% of the hottest hotspots are currently not protected (especially in the tropical regions of Africa), accounting globally for 1.2 million km² of land. Our analysis showed that these areas consist mainly of hotspots for cropland expansion, while 92% of the conflict hotspots for intensification are already under protection (Supplementary Note 6). As previous research found that even more proactive conservation schemes (e.g. Last of the Wild) may overlook many at-risk regions[47], this suggests the need for incorporating future agricultural projections into current conservation prioritisation schemes, in order to protect highly biodiverse but agriculturally desirable areas. We also tested the effect of a policy scenario that would restrict cropland expansion to

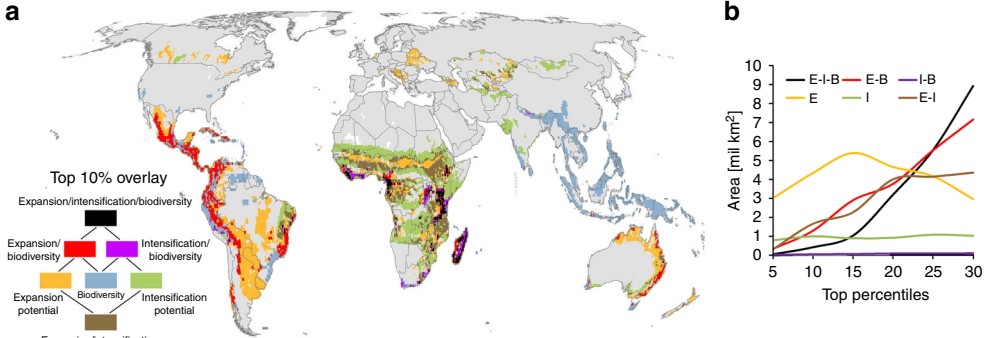

**Fig. 5** Quantile overlay of expansion potential, intensification potential and endemism richness. While **a** shows the 10th percentile, **b** shows the sensitivity of the overlay based on selected thresholds between 5 and 30% (see Supplementary Fig. 18 for maps of the different thresholds). The overlay analysis was performed at 55 km resolution of an equal-area grid. The red areas highlight the hottest hotspots, where high biodiversity may be particularly threatened by future cropland expansion. The purple areas highlight the hottest hotspots, where high biodiversity may be particularly affected by future cropland intensification. The black areas pinpoint places where high biodiversity is particularly threatened by both agricultural scenarios simultaneously. The orange, green and blue colours indicate regions with the top 10% of expansion potential, intensification potential and biodiversity that do not overlap with any other top 10% of data. The brown areas indicate regions where the top percentile of expansion and intensification potential overlap without overlapping the top percentile areas of biodiversity. In **b**, the same colours are used, abbreviating expansion (**E**), intensification (**I**) and biodiversity (**B**)

unprotected areas. While the relative changes of expansion area between the two expansion scenarios are relatively small at global scale, the changes range between +10% and −3% for regions with expansion areas greater 100,000 km². Strongest absolute reductions in the policy scenario which restricts cropland expansion occurred in Sub-Saharan Africa (70,000 km²), Rest of Latin America, and Australia/New Zealand (13,700 km² each). Additional expansion took place in Brazil (76,600 km²) and Canada (10,500 km²) (Supplementary Note 7).

In combination, our results from the economic and biodiversity analyses demonstrated that while cropland expansion will in most cases affect areas important for conservation (regions with the highest production gains in Central and South America (Fig. 2) have significantly high endemism richness (Figs. 3a, 5a), cropland intensification is possible in areas with lower biodiversity (regions with the highest production gains in Sub-Saharan Africa, Central India, Northeast China and Former Soviet Union (Fig. 2) occur in globally less biodiverse regions (Figs. 3b, 5a). These regions largely coincide with the extensive cropping land system archetype[24], where large production gains could be achieved by closing yield gaps through nutrient and water management[36] without necessarily promoting additional decline in biodiversity on the current or future farmlands, e.g. via the use of precision or climate smart agriculture. However, previous studies cautioned against such generalised conclusions about sustainable intensification[35,49,50]. Sub-Saharan Africa and Former Soviet Union are heterogeneous regions that still harbour valuable diversity of species. Even though they are not recognised as biodiversity hotspots globally, largely due to the latitudinal gradient of species richness, many places in Eastern Europe are considered strongholds of agricultural biodiversity on the continent, especially when compared to industrialised farmland in other parts of Europe[51,52]. Therefore, it is likely that the regionally important biodiversity, especially of farmland species, would face the risk of extinction if the extensive forms of farming were replaced by intensive agriculture. This risk would be even exacerbated if agricultural intensification reduced crop genetic diversity, e.g. by encouraging farmers to switch from diverse landraces to hybrids. This, in turn, may reduce field-scale diversity of many taxa in agroecosystems due to a narrower range of food resources and homogenization of crop architecture[53,54]. Again, this shows the need for proactive consideration of different possible farming systems in terms of both expansion or

intensification and more detailed and context-specific analyses that consider also other ecosystem services, such as carbon sinks, and resilience of the land-use systems to conclude whether and how regions could be used for expansion or could be intensified sustainably[55].

Integrated approaches that estimate global land-use change, such as the approach used here, are inherently associated with multiple sources of uncertainties and largely depend on the quality of input data[56–58]. First, although we used the best available determinants of cropland expansion and intensification potentials, uncertainties in global data on land-use and land-use intensity (such as crop yields, harvested area, etc.) at a fine spatial scale remain a major challenge[59,60]. Second, infrastructural, societal, cultural or political aspects that determine accessibility of land (e.g. due to transportation costs, land tenure, traditional or indigenous land or land in failed states) may determine the realisation of estimated agricultural potentials but are not considered due to a lack of global data. Also, all estimated changes especially for cropland intensification assumed that countries have the economic, technological, infrastructural and institutional means to intensify agricultural production, which could be questioned especially in regions like Africa, where we identify large areas with the highest potential for cropland intensification. Third, our models provide cropland expansion and intensification potentials at a 30 arc-second resolution, but the best currently available global measures of biodiversity distribution are not available at such fine scales. The 55-km grid cells are already on the verge of acceptable accuracy because aggregations of species ranges at scales below 2 arc-degrees of resolution may overestimate species richness[61]. Here we alleviated this issue by staying away from a simple measure of species richness and put emphasis on endemism (i.e. range sizes of species within an assemblage), knowing that hotspots of species richness are typically not congruent with endemism or threat[62]. This approach also avoided the utilitarian assumption that landscapes with the most species have the highest conservation value[63]. However, we cannot be certain that the habitats at risk from cropland expansion or intensification in each 55-km grid cell are the same ones in which species occur. For example, hotspot regions in the tropics may have valuable habitats distributed along an elevation gradient but only lowland habitat may be under pressure because topographic, soil and accessibility conditions restrict agricultural suitability in highlands. Nonetheless, our approach allows for exchanging or adding

biodiversity data from other sources, e.g. when more recent or higher-resolution data are available. Fourth, in contrast to biodiversity models, such as GLOBIO that use empirically derived matrix of changes in mean species abundances following a land transformation[64], our approach highlights the main areas at risk, ignoring the mechanisms how expansion or intensification threatens biodiversity. The impacts of continuing land conversion are often non-linear and can vary with spatial configuration[19,65], while indirect effects of intensification threaten biodiversity beyond agricultural lands, due to agrochemical run-offs, habitat homogenization or introduction of invasive species[11,14]. Various aspects of these uncertainties in our integrated approach could be addressed, for example, by applying our method to past data and comparing the identified hotspots with e.g. cropland expansion data derived from remote sensing.

Despite these caveats, our study provides a global perspective of the complex trade-offs between cropland expansion, intensification and biodiversity. Contrasting two scenarios of future production growth clearly demonstrates that each scenario leads to fundamentally different levels and spatial patterns of crop production and prices as well as distributions of the most at-risk areas. Arguably, it is unrealistic to assume that identified hotspot regions will curtail cropland expansion or intensification when there are pressing needs for food or income[65]. However, it can be realistically assumed that best management practices implemented locally or sustainable goals coordinated internationally will help harmonising biodiversity conservation and agricultural production[19].

Our results also provide global-scale spatially explicit contribution to the still unresolved debate on land sharing vs. land sparing[66–68]. Similar to the social-ecological systems model approach[69], we move forward from the bipolar framework by treating agricultural landscapes as complex social-ecological systems, accounting for socio–economic aspects of food production, and stressing the conservation value of biodiversity. Assuming a global land sparing approach, regions where high agricultural potentials were associated with low levels of endemism richness (orange High–Low clusters in Fig. 3) may be suitable for increased crop production at relatively small trade-offs with biodiversity compared to other regions, which could open up the scope for sparing in regions with biodiversity hotspots that would be otherwise threatened by agricultural pressure in the near future.

Even though land-use decisions are made at much finer scales, we identified global hotspots where the debate is most relevant and where additional studies should investigate on a more regional to local level[70]. Because global-scale results are rarely directly transferable to finer spatial scales[71], this should be done by employing regional biodiversity data and downscaled economic analyses, although with the drawback that regional CGE models are limited in considering bilateral trade flows. Moreover, policy decisions aiming at harmonizing agricultural production and conservation, such as land conversion zoning or financial incentives, will have to consider also non-provisioning ecosystem services, rural development objectives, and regional cultural conditions, as well as social and economic implications of, e.g. different strategies for intensification[72]. At the same time, stable governance and effective international organisations are needed to support the implementation of sustainable agricultural strategies because smart land management is a key lever to achieve multiple Sustainable Development Goals[20]. However, identifying the hotspots where future conflicts between biodiversity and agriculture are likely to arise is a first essential step to aid sustainability policies and conservation prioritization schemes. This is also becoming increasingly important with regard to efforts for increasing future bioenergy demand[73] and negative emission scenarios, by use of bio-energy carbon capture and storage (BECCS) technologies[74]. Integrative approaches, such as the one presented here, support the calls for assessing the trade-offs in alternative agricultural pathways and can ultimately help us to meet production goals while maintaining our vital life-support systems.

## Methods
We iteratively link a global CGE model and a dynamic crop growth model to determine the impacts of cropland expansion and intensification on agricultural markets. Cropland expansion and intensification potentials for 2030 are used to identify spatial associations with areas of high biodiversity value (Fig. 1).

**Integrated agricultural potentials.** Our integrated approach combined biophysical and socio–economic conditions to create maps of future agricultural expansion and intensification potentials at 30 arc-sec resolution. The biophysical data covered the period 2011–2040 and considered climate change; the time horizon of the socio–economic data was 2007–2030. The biophysical expansion potential[29] was determined by combining a crop suitability approach for 17 economically important staple and energy crops[37] (Supplementary Table 1) with land availability for cropland expansion, which included all suitable land that is not yet under cultivation[1] or urbanised[75]. We integrated FAO forecasts on expansion[76], that also consider regional socio–economic condition, by using them to weight the biophysical expansion potentials. For details, see Supplementary Note 2.

In case of intensification, biophysical potential yields for the 17 crops were globally simulated on today's cropland[1] using the crop growth model PROMET[6,77]. The ratio between biophysical potential yields and statistical yields[22] resulted in a biophysical intensification potential. These were combined with the marginal profitability of crops that depend on socio–economic scenarios (Supplementary Note 1) that were implemented into the computable general equilibrium model DART-BIO to allocate crops by maximising profit[6]. This resulted in integrated intensification potentials that were used to feed back to the DART-BIO model in terms of changed agricultural productivities, which in turn altered the marginal profitability of crops, such that the re-allocation was repeated iteratively until a stable allocation was established. The integrated model coupling approach allowed to account for changes in land allocation of crops over time due to changing cropping decisions of farmers that depend on changing demand (e.g. population growth, food consumption behaviour) and supply (e.g. climate change, technological progress). For details, see Supplementary Note 3.

**Impact on agricultural markets.** We extracted the top 10% of areas with the highest integrated expansion and intensification potentials to create comparable scenarios of future agricultural pathways. The expansion scenario allowed 7.3 million km² of land to be converted into farmland in places with the highest 10% of expansion potential. The intensification scenario allowed closing yield gaps on 10% of land with the highest intensification potential, up to a level that resulted in the same global production gain. We applied both scenarios in the CGE model DART-BIO to quantify their impact on agricultural markets in terms of crop production, prices, trade and consumption. CGE models solve for balance between supply and demand with flexible prices. For reference, we compared the impact to a 2030 reference scenario that carried forward current trends in demographic growth, gross domestic products and trade policies taking into account that with higher incomes preferences change towards e.g. livestock products. For details, see Supplementary Note 4.

**Agriculture–biodiversity hotspots.** For biodiversity data, we used global range maps for 19,978 species of birds, mammals and amphibians derived from the International Union for Conservation of Nature[78] and Birdlife databases[79]. From these maps, we calculated endemism richness (sum of the inverse extents of occurrence of all species present in a grid cell) because, unlike other biodiversity measures, endemism richness indicates the relative importance of a landscape for conservation by combining aspects of species richness and geographic range size[80]. The data were aggregated in an equal-area grid of 55 × 55 km to provide sufficient detail for global analysis but limit excessive false-presence errors that occur at aggregations of range maps at resolutions below 2 arc-degrees[61]. The hotspots where global biodiversity could be most affected by near-future farmland expansion and intensification were analysed using Local Indicator of Spatial Association (LISA) and quantile overlay. For details, see Supplementary Note 5.

**Reporting summary.** Further information on research design is available in the Nature Research Reporting Summary linked to this article.

## Data availability

Raw datasets analysed in this study are publicly available from the sources provided in Supplementary Note 8 in the supplementary information. All relevant data generated during the study are available upon request from the authors.

## Code availability

The code of the DART-BIO model and for coupling DART-BIO and PROMET is available upon request. PROMET code is not publically available.

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

## Acknowledgements

This project was supported by the German Federal Ministry of Education and Research (grant 01LL0901A: Global Assessment of Land Use Dynamics, Greenhouse Gas Emissions and Ecosystem Services—GLUES and grant 031B0230A: BioNex—The Future of the Biomass Nexus) and the European Structural and Investments Funds (grant CZ.02.1.01/0.0/0.0/16_019/0000797: SustES—Adaptation strategies for sustainable ecosystem services and food security under adverse environmental conditions). We thank Lukas Egli and Carsten Meyer for providing data on species habitat preferences.

## Author contributions

T.V., R.D., F.Z. and R.S. conceived the study and designed the research. R.D., F.Z., T.V. and J.S. collected and analysed the data. R.D. conducted the DART-BIO simulations. T. V. conducted the LISA and biodiversity overlay analysis. F.Z., J.S., W.M. conducted the PROMET simulations and the suitability analysis. R.D. and F.Z. conducted the model coupling between DART-BIO and PROMET. T.V., R.D., F.Z., J.S. and R.S. prepared the manuscript. All authors discussed the results and commented on the paper.

## Additional information

**Competing interests:** The authors declare no competing interests.

