## [Peer Review File · Nature Communications]

Reviewers' comments:

Reviewer #1 - Expertise: Dynamic crop modeling

Dear authors,

in your very comprehensive and detailed study you apply an interdisciplinary approach on the global scale. By coupling a biophysical and socio-economic model and integrating biodiversity data, the impacts of cropland expansion or land-use intensification on agricultural markets and biodiversity are evaluated in a scenario analysis. Your paper is very well written.

In my opinion, the outstanding feature of the work is the presented methodology of coupling complex models and of integrating various environmental and socio-economic data on the global scale for the scenario analysis. It is well demonstrated how a challenging research question can be handled on the global scale. The hotspot analysis is a simple but suitable and effective tool for the integrating quantitative evaluation.

For me, the presented results are maybe more of an exemplary character. The main results which are included in the abstract are quite obvious. And as usual in such complex modelling and in a scenario analysis, the detailed quantitative result data are error-prone due to input data quality and maybe fault assumptions. But that is not a drawback of the study. In the discussion, you mention the multiple sources of uncertainties. This the reader has to bear in mind when studying the presented results.

Maybe it would be a good idea to even more clearly mention that this study is strongly based on the papers by Delzeit, R., Zabel, F., Meyer, C. & Václavík, T. Addressing future trade-offs between biodiversity and cropland expansion to improve food security. *Regional Environmental Change* 17, 1429-1441, doi:10.1007/s10113-016-0927-1 (2017) and by Mauser, W. et al. Global biomass production potentials exceed expected future demand without the need for cropland expansion. *Nat Commun* 6, doi:10.1038/ncomms9946 (2015). The improvements and extensions of the new study should be more clearly stated.

My suggestion for the title:

Future global farming: Trade-offs between cropland expansion, land use intensification and

In the text, maybe you could use abbreviations for the two terms "cropland expansion" and "land use intensification".

Keywords: substitute "expansion potentials" and add "biodiversity"

The abstract and introduction are appropriate. Please add "Conclusions".

The first paragraph of the discussion is too similar to the introduction. In the second paragraph, there are too many repetitions.

Please try to shorten the discussion, but consider these papers:

Henry RC, Engström K, Olin S, Alexander P, Arneth A, Rounsevell MDA (2018) Food supply and bioenergy production within the global cropland planetary boundary. *PLoS ONE* 13(3): e0194695. <https://doi.org/10.1371/journal.pone.0194695>

Lucas A. Garibaldi, Barbara Gemmill-Herren, Raffaele D'Annolfo, Benjamin E. Graeb, Saul A. Cunningham, Tom D. Breeze, Farming Approaches for Greater Biodiversity, Livelihoods, and Food Security, *Trends in Ecology & Evolution*, Volume 32, Issue 1, 2017, Pages 68-80,

<https://doi.org/10.1016/j.tree.2016.10.001>.

Joern Fischer, David J. Abson, Arvid Bergsten, Neil French Collier, Ine Dorresteyn, Jan Hanspach, Kristoffer Hylander, Jannik Schultner, Feyera Senbeta, Reframing the Food–Biodiversity Challenge, *Trends in Ecology & Evolution*, Volume 32, Issue 5, 2017, Pages 335-345,

<https://doi.org/10.1016/j.tree.2017.02.009>.

Alison R. Holt, Anne Alix, Anne Thompson, Lorraine Maltby, Food production, ecosystem services and biodiversity: We can't have it all everywhere, *Science of The Total Environment*, Volume 573, 2016, Pages 1422-1429, <https://doi.org/10.1016/j.scitotenv.2016.07.139>.

The quality of presentation:

Many figures and tables (mainly in the Supplementary Information) need to be improved.

Can error bars in the graphs be added?

What about considering trends in the livestock sector? The linkage between livestock production and the crop sector is extremely strong. Livestock production is the largest user of agricultural land.

In my opinion, the list of considered crops for the modelling is not appropriate since it is not representative for the global scale. It seems as if a list of important crops for central Europe only was a bit extended. What about upland rice as a very important staple food (not only paddy rice)? And why are cereals like rye listed? On a global scale, even oat is more important than rye. And why the distinction in winter wheat and summer wheat? Are these really the most economically important staple and energy crops?

Adjusting the phenological development of the crops to the meteorological conditions is of course a great simplification. In reality, there is a choice of cultivars for each crop with different phenological traits. However, the decision makers in agriculture are slow adapting to new cultivars. It would be interesting how yield would be affected by changed phenological phases.

You considered climate change in the meteorological model input data. Did you also consider an increased atmospheric CO₂ concentration? Can the different effects of increased CO₂ concentration on C₄ and C₃ plants be simulated?

When analysing the intensification potential, are the costs of more needed fertilizer considered? When e.g. the number of harvests per year is increased, the soil will need more fertilizer. And the costs for irrigation?

If I'm right, a validation analysis of your overall results e.g. for the year 2016 would be possible? For selected areas, one could compare the integrated model results with yield and land use statistics? By analysing satellite data, one could derive the expansion of crop areas? This of course can't be covered in one paper, but you can add a sentence mentioning this possibility (or future outlook).

Is it possible to breakdown your study to smaller regions with another CGE model? Would it be possible to transfer your developed methods to e.g. East Africa? Or to Great Britain (a Brexit and a non-Brexit scenario)? Well, this is beyond the scope! I perfectly understand that your study is on the global scale, but for deriving strategies for biodiversity conservation it would be nice to have study regions e.g. on the continental or on a smaller scale. Maybe you could add a sentence about the transferability of your integrated modelling approach to other spatial scales.

In your study, you considered the biodiversity regarding birds, mammals and amphibians. It would be very interesting to (instead or in addition) consider insects and plants. May I cite some text from the

website of the PREDICTS project (<https://www.predicts.org.uk/>):

"The project aims to include data from a wide variety of different taxonomic groups, including vertebrates, invertebrates and plants. Indicators of biodiversity declines have been heavily biased towards taxonomic groups that appeal to many naturalists, such as birds. Different taxonomic groups may respond differently to anthropogenic pressure, and it is important to include as wide a range of species when considering global changes in biodiversity in response to human impact."

Would data from the Global Biodiversity Information Facility (<https://www.gbif.org/>) be suitable?

It is a great advantage of your research approach that you can (I suppose more or less easily) exchange or add biodiversity data to analyse a whole set of questions regarding the trade-offs between agriculture and biodiversity. Please make this advantage very clear!

The dynamic crop growth model should be added in Fig. 1. The "Biophysical constraints" and "Socio-economic conditions" could only appear once, marked as model input data.

Would it be possible to shorten the Supplementary Information?

A few small remarks:

Line 14/15: benefited in which context?

18: India and China

25: Introduction

43/44: are there newer papers to be cited?

add a sentence explaining land use intensification in the introduction

Fig. 1: line 88: global cropland expansion

line 89-92: this is not a sentence

113: net-importer

Fig 2: revise the terms in the legend and the order of the terms of the y-axis (don't use "Globe") and don't list "Rest of the World" as second one. Avoid green and orange when using only two colours.

Maybe shift the y-axis to the centre.

154: 14% and 8%

162: 64% and 66%

164: 41%

206: 5% to 30%

Fig. 4: add information about the brown colour

236: 55 km resolution

246: net-importer in the ... net-exporter in the ...

In the first half of the paper, you speak of farming strategies, in the discussion more of farming pathways. I would stick to the term "strategies".

253: ... resulted in an increase ...

285: this suggests

297: also mention the term "climate-smart-agriculture"?

354-358: this sentence is too long

367: is "leverage" the right term?

372: rephrase "our vital life-support systems"

405: DART-BIO model

Supplementary Information

S1.1

add a sentence about crop management and model validation

S1.2

add a sentence about model validation
line 91/92: ... in 2030 in a cropland ... scenario.

Fig. S4

improve order (e.g. do not start with "Rest of Latin America") and improve graph

line 220 and 318: ...top 10%...

Line 300: according to (27) ...

Fig. S8: symbols are not seen when printed in grey colours

improve order

Former Soviet Union

S4

the text needs to be improved

what are "other oil seeds"? what are "other grains"?

Fig. S9: please don't use green and orange for the two colours

Fig. S10 to S14: shift the legend so that there is enough space for the x-axis to be expanded

don't use green and orange

Fig. S16 and Table S3: change order (it is strange to read "global average" and then "rest of the world")

don't use green and orange

Fig. S18: add description for brown colour

Table S2 and S3: improve layout (numbers right-aligned), do not use points as thousands separator, change terms in the columns (and in the title of Table S3).

Fig. S19: rephrase „LISA expansion area“

Thanks for your interesting research!

Reviewer #2 - Expertise: Agricultural economics, CGE models

This paper uses a simulation approach to model trade-offs between cropland expansion, land use intensification, and biodiversity. Its motivation is the challenge of meeting future global biomass (including food) demands while conserving ecosystems and biodiversity. This is an important research question. The study integrates trade, socio-economic and biophysical modeling in a novel way that makes sense. It finds that cropland expansion and intensification lead to different levels and spatial patterns of crop production and prices as well as pressure on the most at-risk (hotspot) areas. A goal of this analysis is to provide input into policies to harmonize biodiversity conservation and agricultural production to satisfy future global food needs. The focus of my comments is mostly on the economics side.

The CGE modeling in this paper is reminiscent of that used in Hertel et al. (PNAS, 2014) to address similar land use (though different environmental) questions. I think you should cite that and perhaps one or two other papers that use a similar global modeling approach and explain how yours is different (or not).

GTAP models are highly aggregated compared with the biophysical model. A key assumption seems to be that markets within regions convey changes in global prices to the most at-risk localities with respect to biodiversity. Other studies (e.g., by Arslan, Dyer, Taylor, and others) question this, particularly in isolated areas that (I would guess) might be biodiversity hotspots. If food prices rise (fall), barriers to price transmission would seem to be good (bad) news for biodiversity, and your model might overstate (understate) negative impacts. This is important enough that I think you need to deal with it head-on instead of hinting at it in the caveats.

The literature on crop genetic diversity adds another element to the food production-biodiversity relationship. There, a major concern is that agricultural intensification reduces crop genetic diversity, e.g., by encouraging farmers to switch from diverse landraces to hybrids. This element does not seem to be part of your paper. It would be good to mention it and for readers to have some sense (if you can give it) on possible correlations between crop genetic diversity and the biodiversity your study focuses on.

Reviewer #3 - Expertise: Agriculture and land use effects on biodiversity

This article investigates the biodiversity impacts of meeting 2030 food demand by crop expansion or by land-use intensification. This is a very interesting question and the authors have worked hard to generate realistic spatial models of crop expansion/intensification. However, the biodiversity analysis is not strong, and I feel that the result are over-simplified maps.

Major comments

(1) The authors assume that agricultural intensification will affect endemics in many regions. However, the identified areas often have forest endemics that depending on the crop type are already unlikely to reside in any meaningful numbers or at all within farmland. For such species, there is no quantification of how severe the risk is or how many species may be predicted to go extinct following intensification. Rather, 'hotspots' are simply identified by matching areas with crop potential to those with high biodiversity.

(2) "Here we alleviated this issue by staying away from a simple measure of species richness and put emphasis on endemism (i.e. range sizes of species within an assemblage), knowing that hotspots of species richness are typically not congruent with endemism or threat". The authors claim their study makes a big improvement on those that deal with single numbers (eg species richness), but this is pretty much what they have done here just generating a species richness number for endemic species rather than all species. They have ignored how we would expect that number to change with agricultural intensification/expansion.

(3) I could not see where market feedbacks were incorporated in their models. Of particular concern, some areas might be more intensively farmed but this leads to higher profitability and then increased expansion. I.e. there could be a coupled process of intensification AND expansion rather than intensification OR expansion.

(4) For an article designed to identify the impacts of agricultural management on biodiversity, it was disappointing not to see potential policy drivers of change being unpicked. For instance, (i) intensification coupled with land planning/zoning that prevents conversion and (ii) a role of other markets that fund protection vs farming, amongst others.

Reviewer #1 - Expertise: Dynamic crop modeling

Dear authors,

in your very comprehensive and detailed study you apply an interdisciplinary approach on the global scale. By coupling a biophysical and socio-economic model and integrating biodiversity data, the impacts of cropland expansion or land-use intensification on agricultural markets and biodiversity are evaluated in a scenario analysis. Your paper is very well written.

In my opinion, the outstanding feature of the work is the presented methodology of coupling complex models and of integrating various environmental and socio-economic data on the global scale for the scenario analysis. It is well demonstrated how a challenging research question can be handled on the global scale. The hotspot analysis is a simple but suitable and effective tool for the integrating quantitative evaluation.

For me, the presented results are maybe more of an exemplary character. The main results which are included in the abstract are quite obvious. And as usual in such complex modelling and in a scenario analysis, the detailed quantitative result data are error-prone due to input data quality and maybe fault assumptions. But that is not a drawback of the study. In the discussion, you mention the multiple sources of uncertainties. This the reader has to bear in mind when studying the presented results.

Maybe it would be a good idea to even more clearly mention that this study is strongly based on the papers by Delzeit, R., Zabel, F., Meyer, C. & Václavík, T. Addressing future trade-offs between biodiversity and cropland expansion to improve food security. *Regional Environmental Change* 17, 1429-1441, doi:10.1007/s10113-016-0927-1 (2017) and by Mauser, W. et al. Global biomass production potentials exceed expected future demand without the need for cropland expansion. *Nat Commun* 6, doi:10.1038/ncomms9946 (2015). The improvements and extensions of the new study should be more clearly stated.

Response: We now pronounce more clearly that this study is based on the above mentioned publications. We added the following sentence to the introduction:

“First, we combine two established approaches from previous work of the authors^{6, 29}, which integrate both biophysical and socio-economic conditions to...

My suggestion for the title: Future global farming: Trade-offs between cropland expansion, land use intensification ...

Response: “Farming the future planet” was to refer to the earlier papers (Monfreda, Ramankutty et al. 2008, Ramankutty, Evan et al. 2008) entitled with “Farming the planet”, which were the first papers that addressed the current distribution of global cropland. We now provide one of the first outlooks of possible future developments in this respect. As this reference was not clear to the reviewers, we dropped this first part and now have “Global trade-offs between cropland expansion, intensification and biodiversity in the context of agricultural markets” as title.

In the text, maybe you could use abbreviations for the two terms “cropland expansion” and “land use intensification”.

Response: Based on your suggestion we reconsidered the terms “cropland expansion” and “land use intensification” and decided to change “land use intensification” into “cropland intensification” since it is more specific and better explains the focus of our analysis. This also shortens the description in the text. However, we did not follow the suggestion to introduce abbreviations, such as CE or CI, as we feel these are not common sense and would make the paper less accessible by a broader audience. Moreover, these could lead to misinterpretations (e.g. CI could be the abbreviation for ‘cropland intensification’, but it is also commonly used for the NGO ‘Conservation International’ or ‘confidence interval’, both relevant in the field of our paper).

Keywords: substitute “expansion potentials” and add “biodiversity”

Response: Changed as suggested.

The abstract and introduction are appropriate. Please add “Conclusions”.

Response: The journal format does not allow adding the header “Conclusions.” The conclusion is at the end of our discussion, where we describe several conclusions that we draw from our results.

The first paragraph of the discussion is too similar to the introduction. In the second paragraph, there are too many repetitions. Please try to shorten the discussion, but consider these papers: Henry RC, Engström K, Olin S, Alexander P, Arneth A, Rounsevell MDA (2018) Food supply and bioenergy production within the global cropland planetary boundary. PLoS ONE 13(3): e0194695.

Lucas A. Garibaldi, Barbara Gemmill-Herren, Raffaele D’Annolfo, Benjamin E. Graeb, Saul A. Cunningham, Tom D. Breeze, Farming Approaches for Greater Biodiversity, Livelihoods, and Food Security, Trends in Ecology & Evolution, Volume 32, Issue 1, 2017, Pages 68-80, <https://doi.org/10.1016/j.tree.2016.10.001>.

Joern Fischer, David J. Abson, Arvid Bergsten, Neil French Collier, Ine Dorresteijn, Jan Hanspach, Kristoffer Hylander, Jannik Schultner, Feyera Senbeta, Reframing the Food–Biodiversity Challenge, *Trends in Ecology & Evolution*, Volume 32, Issue 5, 2017, Pages 335-345, <https://doi.org/10.1016/j.tree.2017.02.009>.

Alison R. Holt, Anne Alix, Anne Thompson, Lorraine Maltby, Food production, ecosystem services and biodiversity: We can't have it all everywhere, *Science of The Total Environment*, Volume 573, 2016, Pages 1422-1429, <https://doi.org/10.1016/j.scitotenv.2016.07.139>.

Response: We included the references in the discussion where appropriate. We shortened the discussion and removed repetitions and redundancies within the introduction and discussion. We also moved some basic points from the discussion to the introduction; see e.g. the third paragraph of the introduction:

“Changes in agricultural productivity are addressed in some scenario studies feeding yield changes into partial or general equilibrium models, but feedbacks from the economic model to biophysical models are neglected. Thus, emerging trade-offs have not yet been addressed using comparable scenarios that integrate biophysical and socio-economic drivers of crop production.”

The quality of presentation: Many figures and tables (mainly in the Supplementary Information) need to be improved.

Response: We revised and improved the graphic quality of all figures in the main text and the SI and also included maps in high resolution. See detailed comments on figure changes below.

Can error bars in the graphs be added?

Response: On the basis of our models and the resulting data, we cannot provide error bars for figures. For instance, we do not use different climate scenarios or apply various models that one could use for providing error bars across scenarios or models.

What about considering trends in the livestock sector? The linkage between livestock production and the crop sector is extremely strong. Livestock production is the largest user of agricultural land.

Response: We consider trends in the livestock sector in DART-BIO. Our scenarios assume that the current trends in demographic growth, gross domestic products, trade policies are carried forward with it consequences for increasing demand in livestock feed due to higher incomes and increased preferences towards livestock products.

We now mention this briefly in the main text (see Materials and Methods, Section ‘Impact on agricultural markets’): “We compared the impact to a 2030 reference scenario that carried forward current trends in demographic growth, gross domestic products and trade

policies taking into account that with higher incomes preferences change towards e.g. livestock products.“) and added more information on trends in livestock sector in the supplementary information section S1.3.

In my opinion, the list of considered crops for the modelling is not appropriate since it is not representative for the global scale. It seems as if a list of important crops for central Europe only was a bit extended. What about upland rice as a very important staple food (not only paddy rice)? And why are cereals like rye listed? On a global scale, even oat is more important than rye. And why the distinction in winter wheat and summer wheat? Are these really the most economically important staple and energy crops?

Response: The original GTAP database used in the DART-BIO model differentiates 6 crop categories only, which we disaggregated further into 10 crop categories based on information of the Food and Agricultural Organization (see Calzadilla, Delzeit et al. (2016) for a detailed description). These 10 crop categories are represented by 17 crops, which are modelled in PROMET. For better documentation we presented this mapping of crops and crop categories in Table S2 in the supplement.

These 17 crops represent 73% of the global cropland area and 73% of global crop production according to FAOSTAT (all listed crops, average for 1981-2010). With our choice of crops we intend to cover the most important crops with respect to area and economic importance. They not only represent staple crops, but also energy crops, which captures the trends in political support of biofuels. It includes maize, wheat, rice and soy, that provide 2/3 of global calorie production, but also considers more regionally important food crops, such as millet or cassava. Further, we capture the main bioenergy crops, such as oil palm, maize, soy, sugarcane and rapeseed that largely drive land use change in many regions.

As this might have cut too short in our paper, we added the following sentence to the last paragraph of the introduction of the main paper:

“These crops represent 73% of global cropland area and crop production and cover the most important staple and energy crops, to also capture trends in political support of biofuels.”

Although upland rice may be an important staple crop in specific regions (e.g. highlands of Vietnam or the Philippines), we did not include it in our list because traditional rice varieties are being rapidly replaced in most regions by high-yielding varieties. This is because seed is by far the most expensive input in those traditional regions and, more importantly, the yields of traditional upland rice are quite low and thus serve as direct subsistence for farmers, not being sold as a commodity on the market (Settele, Heong et al. 2018, Spangenberg, Beaurepaire et al. 2018).

Regarding the question, why rye is listed: even though there are crops with larger area or production than rye, we used rye as an additional crop to represent the crop category GRON (rest of cereal grains, compare Table S2). The crop parameterization in PROMET was already available.

It is required to distinguish between summer and winter wheat in the crop model, since these crops differ largely both in phenology and physiology, which results in different yields.

Adjusting the phenological development of the crops to the meteorological conditions is of course a great simplification. In reality, there is a choice of cultivars for each crop with different phenological traits. However, the decision makers in agriculture are slow adapting to new cultivars. It would be interesting how yield would be affected by changed phenological phases.

Response: The phenological adjustment represents the selection of a crop adapted to the local meteorological conditions. This is applied not annually, but for a 30-year average, assuming that the farmer selects the cultivar on basis of his 30-year practical experience. The same methodology is also used e.g. in the Global Gridded Crop Model Initiative (GGCMI) phase II protocol, that is part of the Agricultural Model Intercomparison and Improvement Project (AGMIP) for adapting crops to changed climate conditions (e.g. Zabel, Hank et al. 2019). The impacts of adapting crop phenology to increased temperatures on yields are currently investigated for various global crop models that participate in GGCMI phase II (PROMET included). Related papers have already been submitted but are not yet published. We are grateful for that input and added this aspect briefly to the supplement (S3.1):

“We assumed that cultivars are adapted to the changed temperature conditions in 2011-2040 by adjusting phenological speed for each location to the same length between harvest and maturity as in the reference baseline simulation from 1981-2010. The adjustment takes place for climate averages and not annually, since we assume that farmers select cultivars on basis of a long-term practical experience, and thus are not immediately adapting new cultivars.”

You considered climate change in the meteorological model input data. Did you also consider an increased atmospheric CO₂ concentration? Can the different effects of increased CO₂ concentration on C₄ and C₃ plants be simulated?

Response: Yes, we consider the increase in atmospheric CO₂ concentrations. The PROMET model considers the C₃ and C₄ pathways for the respective crops (see Supplement, Section S1.1).

When analysing the intensification potential, are the costs of more needed fertilizer considered? When e.g. the number of harvests per year is increased, the soil will need more fertilizer. And the costs for irrigation?

Response: For both options, expansion and intensification, no additional costs are considered in DART-BIO. This is for two reasons: 1) We do not aim at performing a welfare analysis, which would imply to consider these costs since otherwise the producer rent would be overestimated.

2) There is no global data available on expansion costs; hence for consistency and comparability of the two scenarios we assume no additional costs for both options. Therefore, we only talk about changes in production and prices. Thanks for pointing us out to this. We address this in the first paragraph of the discussion:

“For consistency, we assumed neither costs of expanding cropland nor costs for intensifying production.”

If I'm right, a validation analysis of your overall results e.g. for the year 2016 would be possible? For selected areas, one could compare the integrated model results with yield and land use statistics? By analysing satellite data, one could derive the expansion of crop areas? This of course can't be covered in one paper, but you can add a sentence mentioning this possibility (or future outlook).

Response: Many thanks for this input and suggestion. Validation is truly a crucial issue in the entire field of global (scenario) studies like ours. While we base our analysis on well validated models, our results refer to a future scenario which nobody knows. Especially distinguishing the different strengths of effects of intensification and expansion are hard to validate, as those are overlaid in all real-world observations. However, for future outlook, it could be possible to apply our method for past data and compare this as suggested e.g. with expansion data derived from remote sensing data. Nevertheless, validating agricultural potentials is tricky and it is not yet clear if appropriate and consistent data are available at the global scale. As suggested, we added a short text on this topic to the sixth paragraph of the discussion:

“Various aspects of these uncertainties in our integrated approach could be addresses, for example, by applying our method to past data and comparing the identified hotspots with e.g. cropland expansion data derived from remote sensing.”

Is it possible to breakdown your study to smaller regions with another CGE model? Would it be possible to transfer your developed methods to e.g. East Africa? Or to Great Britain (a Brexit and a non-Brexit scenario)? Well, this is beyond the scope! I perfectly understand that your study is on the global scale, but for deriving strategies for biodiversity conservation it would be nice to have study regions e.g. on the continental or on a smaller scale. Maybe you could add a sentence about the transferability of your integrated modelling approach to other spatial scales.

Response: Yes, it is principally possible to do such a study at finer spatial scale. PROMET already runs at fine spatial scale and is aggregated to match with DART-BIO in order to couple the models. By the use of national CGE models, a more regional analysis would be possible. The drawback of using national CGE models would be that bilateral trade flows could not be taken into account. According to our conclusions, the global results of our study could be used to conduct regional studies systematically in the hotspot regions in order to better investigate regional effects. Therefore, the global economic trends need to be considered, that could be provided by our global study to be consistent and comparable between the different regional studies (Vaclavik, Langerwisch et al. 2016).

As suggested, we added the following text to the last paragraph of the discussion:

“Because global-scale results are rarely directly transferable to finer spatial scales⁷¹, this should be done by employing regional biodiversity data and downscaled economic analyses, although with the drawback that regional CGE models are limited in considering bilateral trade flows.”

In your study, you considered the biodiversity regarding birds, mammals and amphibians. It would be very interesting to (instead or in addition) consider insects and plants. May I cite some text from the website of the PREDICTS project (<https://www.predicts.org.uk/>): "The project aims to include data from a wide variety of different taxonomic groups, including vertebrates, invertebrates and plants. Indicators of biodiversity declines have been heavily biased towards taxonomic groups that appeal to many naturalists, such as birds. Different taxonomic groups may respond differently to anthropogenic pressure, and it is important to include as wide a range of species when considering global changes in biodiversity in response to human impact." Would data from the Global Biodiversity Information Facility (<https://www.gbif.org/>) be suitable? It is a great advantage of your research approach that you can (I suppose more or less easily) exchange or add biodiversity data to analyse a whole set of questions regarding the trade-offs between agriculture and biodiversity. Please make this advantage very clear!

Response: In our study, we used IUCN and Birdlife data on mammals, birds and amphibians because these are currently the only available spatially explicit data on global biodiversity. For this reason, recent global-scale studies with similar aim also relied on data for these taxonomic groups. The GBIF and especially the PREDICTS databases are great sources of data collated for many taxa with the aim to quantify species and community responses to a range of anthropogenic pressures. While the data can be used to roughly estimate biodiversity loss per land use change class, the spatial coverage is insufficient for our purpose. For example, many regions, such as Sub-Saharan and Central Africa, Central Asia, or Eastern Europe are severely under-represented in the databases (please see map of spatial coverage of biodiversity measurements in PREDICTS: <https://www.predicts.org.uk/pages/outputs.html>).

However, to account for different responses of different species, we have now extended our analysis to address the reviewers concerns and included additional analyses that distinguish

between forest and natural habitat specialists versus species that are regular or at least marginal cropland users. Please, see our detailed response to Reviewer 3.

In addition, we now acknowledge in the paper that the advantage of our approach is that it allows to exchange or add biodiversity data to examine more specific questions on the trade-offs between agriculture and biodiversity, which is in accordance to your comment. See paragraph six of the discussion section of the main paper:

“Nonetheless, our approach allows for exchanging or adding biodiversity data from other sources, e.g. when more recent or higher-resolution data are available.”

The dynamic crop growth model should be added in Fig. 1. The “Biophysical constraints” and “Socio-economic conditions” could only appear once, marked as model input data.

Response: We added labels for the main method/model for each stream of data in Fig. 1.: “agricultural suitability model” for expansion potential and “dynamic crop growth model” for intensification potential. We kept biophysical constraints and socio-economic conditions separately to make it clear that they are separate inputs into the two different models. But we also added “IUCN data on species ranges” for biodiversity and changed “baseline” scenario to “reference” scenario to keep it consistent with the text.

Would it be possible to shorten the Supplementary Information?

Response: Since Supplementary Information is not limited in terms of space, we wanted to provide complete and full account of our data, methods and models to ensure transparency and reproducibility of our study. This is important especially because our study integrates three different and complex analyses from different disciplines. Additionally, some of the information can be found in earlier papers. However, our aim was to provide all information needed to fully understand and at best redo the analysis, which resulted in a quite large supplementary. If the editor requests shortening the supplement, we of course can remove some parts.

A few small remarks:

Main text:

Line 14/15: benefited in which context?

We revised the sentence, referring now to beneficial effects on food security.

18: India and China

Corrected

25: Introduction

Corrected

43/44: are there newer papers to be cited?

Yes, we included two more recent papers (Siebert, Kummu et al. 2015, Müller, Elliott et al. 2016).

Add a sentence explaining land use intensification in the introduction

Response: In the second paragraph of the introduction, we describe possible measures for intensification that are supposed to have negative impacts on biodiversity. We added also examples for agro-chemicals.

Fig. 1: line 88: global cropland expansion

Corrected

line 89-92: this is not a sentence

Corrected

113: net-importer

Corrected

Fig 2: revise the terms in the legend and the order of the terms of the y-axis (don't use "Globe") and don't list "Rest of the World" as second one. Avoid green and orange when using only two colours. Maybe shift the y-axis to the centre

Response: As suggested, we re-ordered the terms of the y-axis and revised the terms in the legend. Why should green and orange be avoided? As far as we know, green and orange can be differentiated by colour-blind people. The colours have been chosen to match the symbology of the respective scenarios in global maps (Quantile overlays Fig 5 and Fig S18).

154: 14% and 8%

Corrected

162: 64% and 66%

Corrected

164: 41%

Corrected

206: 5% to 30%

Corrected

Fig. 4: add information about the brown colour

Added

236: 55 km resolution

Corrected

246: net-importer in the ... net-exporter in the ...

Corrected

In the first half of the paper, you speak of farming strategies, in the discussion more of farming pathways. I would stick to the term “strategies”.

Corrected

253: ... resulted in an increase ...

Corrected

285: this suggests

Corrected

297: also mention the term “climate-smart-agriculture”?

Corrected

354-358: this sentence is too long

Sentences split, revised.

367: is “leverage” the right term?

Corrected

372: rephrase “our vital life-support systems”

Response. We kept the phrase as this was exactly what we aimed at expressing.

405: DART-BIO model

Corrected

Supplementary Information:

S1.1 add a sentence about crop management and model validation

Response: We included to S.3.1 a sentence on our assumptions on crop management and on model validation:

“For identifying intensification potentials (Fig. S5) on existing cropland, we simulated potential yields, assuming a perfect management of crops. This implies that no nutrient stress, pests and diseases occur during crop growth and that optimal sowing dates and potential number of harvests per year, adapted to the climate conditions for 2011-2040, are

applied from globally derived data on the optimal start of the growing season from Zabel, et al., for each of the 17 considered crops (Table S1).”

“The model has been validated at field scale, and simulated global yields have been compared against statistical values and statistical models on a country level.”

S1.2 add a sentence about model validation

Response: We included to S.1.2 a paragraph on model validation:

“CGE models are rarely validated. Van Dijk, et al. state that “Curiously, however, in contrast to modelling efforts in, for example, the biophysical sciences, CGE model findings are seldom subjected to any systematic validation procedure. A cursory review of the literature reveals isolated single country CGE model validation exercises, although with a dearth of available data, there is a paucity of equivalent studies which implement such a procedure in a global CGE context.” While there is not systematic validation procedure performed for the DART-BIO model, the results of the reference run are compared to other business as usual studies such as the OECD/FAO agricultural outlook. Further, the modelling team is engaged in an initiative by the Global Trade Analysis Project that aims to develop best practices for baseline generation.”

line 91/92: ... in 2030 in a cropland ... scenario.

To our understanding, both, ‘in’ and ‘under’ are possible.

Fig. S4 improve order (e.g. do not start with “Rest of Latin America”) and improve graph

Response: The graph was sorted by the share of relative expansion, we reversed the sorting.

line 220 and 318: ...top 10%...

Corrected

Line 300: according to (27) ...?

Corrected

Fig. S8: symbols are not seen when printed in grey colours, improve order

Response: regions are now sorted by average yield gap, colors are removed.

S4: the text needs to be improved

Done

what are “other oil seeds”? what are “other grains”?

Response: We changed the name and added an explanation to the supplementary information.

Fig. S9: please don’t use green and orange for the two colours

See above

Fig. S10 to S14: shift the legend so that there is enough space for the x-axis to be expanded
don't use green and orange

Done

Fig. S16 and Table S3: change order (it is strange to read “global average” and then “rest of the world”)

Done

don't use green and orange

See above

Fig. S18: add description for brown colour

Corrected

Table S2 and S3: improve layout (numbers right-aligned), do not use points as thousands separator, change terms in the columns (and in the title of Table S3).

Corrected

Fig. S19: rephrase „LISA expansion area“

Corrected

Thanks for your interesting research! **Thank you!**

Reviewer #2 - Expertise: Agricultural economics, CGE models

This paper uses a simulation approach to model trade-offs between cropland expansion, land use intensification, and biodiversity. Its motivation is the challenge of meeting future global biomass (including food) demands while conserving ecosystems and biodiversity. This is an important research question. The study integrates trade, socio-economic and biophysical modeling in a novel way that makes sense. It finds that cropland expansion and intensification lead to different levels and spatial patterns of crop production and prices as well as pressure on the most at-risk (hotspot) areas. A goal of this analysis is to provide input into policies to harmonize biodiversity conservation and agricultural production to satisfy future global food needs. The focus of my comments is mostly on the economics side.

The CGE modeling in this paper is reminiscent of that used in Hertel et al. (PNAS, 2014) to address similar land use (though different environmental) questions. I think you should cite that and perhaps one or two other papers that use a similar global modeling approach and explain how yours is different (or not).

Response: Hertel et al. (2014) used the model SIMPLE (Simplified International Model of Prices Land Use and the Environment Mode), which is a simplified partial equilibrium mode (3 sectors). They simulate historic trends by exogenously specifying e.g. total factor productivity (impacting on e.g. land input per output) to address the question whether agricultural innovations are land and CO₂ emission sparing. To put our results in context we added the suggested paper as well as another study by Hertel (Baldos and Hertel 2014), as reference to the introduction.

GTAP models are highly aggregated compared with the biophysical model. A key assumption seems to be that markets within regions convey changes in global prices to the most at-risk localities with respect to biodiversity. Other studies (e.g., by Arslan, Dyer, Taylor, and others) question this, particularly in isolated areas that (I would guess) might be biodiversity hotspots. If food prices rise (fall), barriers to price transmission would seem to be good (bad) news for biodiversity, and your model might overstate (understate) negative impacts. This is important enough that I think you need to deal with it head-on instead of hinting at it in the caveats.

Response: We agree, that GTAP and related global models such as DART-BIO are highly aggregated with respect to spatial resolution as well as crop species considered. To break down spatial effects of cropland expansion (and intensification) we implemented a new approach (first developed for cropland expansion in Delzeit, Zabel et al. (2017)). While price signals are generated for the economies and the 10 crop categories represented in DART-BIO, and thus there is one change in prices for a region (e.g. Brazil), relative price changes impact on the allocation of crops utilizing the integrated intensification potential (of already managed land) for the different crop categories (compare Table S2). The integrated intensification potential is available at a fine spatial resolution (compare Fig. S6). Yet, our analysis is not based on the assumption that global price changes affect most at-risk locations within a region (e.g. Brazil) with respect to biodiversity. Hence, the change in allocation of crops as well as of pasture land is affected by relative prices as well as biophysical characteristics. But price signals do not determine the spatial location of expansion into unmanaged land.

The spatial allocation of the expansion potential (into unmanaged land) is determined by the CGE model *and* the integrated expansion potential which considers 17 crops (see S3.2 and Table S2) including high resolution crop suitability map.

The literature on crop genetic diversity adds another element to the food production-biodiversity relationship. There, a major concern is that agricultural intensification reduces crop genetic diversity, e.g., by encouraging farmers to switch from diverse landraces to hybrids. This element does not seem to be part of your paper. It would be good to mention it and for readers to have some sense (if you can give it) on possible correlations between crop genetic diversity and the biodiversity your study focuses on.

Response: Our analyses did not directly account for the element of crop genetic diversity but we surely agree that it is an important mechanism through which agricultural intensification may affect biodiversity. Therefore, we revised the text in the Discussion as follows:

“Therefore, it is likely that the regionally important biodiversity would face the risk of extinction if the extensive forms of farming were replaced by intensive agriculture. This risk would be even exacerbated if agricultural intensification reduced crop genetic diversity, e.g. by encouraging farmers to switch from diverse landraces to hybrids. This, in turn, may reduce field-scale diversity of many taxa in agroecosystems due to a narrower range of food resources and homogenization of crop architecture.”

Reviewer #3 - Expertise: Agriculture and land use effects on biodiversity

This article investigates the biodiversity impacts of meeting 2030 food demand by crop expansion or by land-use intensification. This is a very interesting question and the authors have worked hard to generate realistic spatial models of crop expansion/intensification. However, the biodiversity analysis is not strong, and I feel that the result are over-simplified maps.

Major comments

(1) The authors assume that agricultural intensification will affect endemics in many regions. However, the identified areas often have forest endemics that depending on the crop type are already unlikely to reside in any meaningful numbers or at all within farmland. For such species, there is no quantification of how severe the risk is or how many species may be predicted to go extinct following intensification. Rather, ‘hotspots’ are simply identified by matching areas with crop potential to those with high biodiversity.

Response: We assume that agricultural intensification affects all species (not only endemics) in regions with high intensification potential, and there is sufficient evidence that describes potential mechanisms of threat, e.g. through habitat homogenization, irrigation or high inputs of agrochemicals (Geiger, Bengtsson et al. 2010, Meehan, Werling et al. 2011, De Frutos, Olea et al. 2015, Beckmann, Gerstner et al. 2019). However, even though studies argue that not only species in cropland but also in surrounding natural habitats are affected, we have very little knowledge about precise ecological responses to different levels of intensification. For example, previous studies that tried to quantify biodiversity loss explored the full option space of possible agricultural change and generated a random response function from within the space of possible responses, yielding extremely wide ranges of biodiversity outcomes (Kehoe, Romero-Muñoz et al. 2017, Egli, Meyer et al. 2018). Due to these severe uncertainties, we restrain from estimating numbers of species that may go extinct and rather focus on spatial patterns of threatened endemism-scaled

richness, i.e. areas with high conservation values. The quantification of biodiversity loss would be additionally uncertain here because we do not assume that yield gaps can be fully closed but rather that in croplands with the highest intensification potential the yield gap is closed up to the level that leads to equivalent production gain in both scenarios.

However, we surely agree that species with different habitat requirements (e.g. forest endemics) are unlikely to reside in large numbers within farmlands and that intensification will not affect all considered species equally. Therefore, we now include additional analyses in the main paper that combine extent-of-occurrence range maps with information on species preferred habitat types. Specifically, we distinguished two groups of species, (1) forest and natural habitat specialists versus (2) regular or at least marginal cropland users, and compared the results of our hotspot analyses for these two habitat groups, in addition to the original three taxa. In summary, intensification affected cropland users more than species in forests and natural habitats but the overall global patterns of conflict hotspots remained quite similar for both groups, suggesting that areas with high biodiversity host large numbers of forest/natural habitat specialists as well as cropland users (see third paragraph in the results section of the main paper and new Figure 4).

(2) “Here we alleviated this issue by staying away from a simple measure of species richness and put emphasis on endemism (i.e. range sizes of species within an assemblage), knowing that hotspots of species richness are typically not congruent with endemism or threat”. The authors claim their study makes a big improvement on those that deal with single numbers (eg species richness), but this is pretty much what they have done here just generating a species richness number for endemic species rather than all species. They have ignored how we would expect that number to change with agricultural intensification/expansion.

Reponse: We are sorry that the description what our biodiversity metric represents was not clear enough. We thus aimed at improving clarity by changing text. Endemism richness sensu (Kier, Kreft et al. 2009) is not a simple species richness number for endemics (as opposed to all species). This indicator does consider all species at a site and is calculated as the sum of the inverse global range sizes of all species present in a grid cell. Therefore, the measure combines species richness with the size of species geographic range and thus indicates the specific contribution of an area (grid cell) to global biodiversity. As the detailed description of this measure was only at the end of the article in Methods, we now clarify its meaning also in the caption of Figure 1:

“Endemism richness integrated IUCN range maps of 19,978 species of mammals, birds and amphibians into a global biodiversity metric aggregated at 55-km resolution of an equal-area grid. This indicator combines species richness with a measure of endemism (i.e. the range sizes of species within an assemblage) and thus indicates the relative importance of a site for conservation.”

We believe using this indicator is more informative than the typical measure of the number of species present in a grid cell and avoids the utilitarian assumption that landscapes with the most species have the highest conservation value.

(3) I could not see where market feedbacks were incorporated in their models. Of particular concern, some areas might be more intensively farmed but this leads to higher profitability and then increased expansion. I.e. there could be a coupled process of intensification AND expansion rather than intensification OR expansion.

Response: Many thanks for pointing out this ambiguity and lack of clarity. We carefully revised the section S.1.2 and S.1.3 of the CGE model to make this clear for a broader community. DART-BIO incorporates feedbacks between different commodity markets.

In DART-BIO, the mobility of land between different economic uses (crops, pasture, managed forest) is restricted by a Constant Elasticity of Transformation function (S1.3). If e.g. the price of soybeans increases, farmers allocate more land to produce soybeans at e.g. the cost of maize. Hence, there is land use change amongst land that is managed / economically used. In the intensification scenario, the productivity of one crop might increase stronger than that of another, hence, farmers produce more of the crops with increased productivity: higher productivity (intensification scenario) results in higher returns to land, and more production of crops. The market effect is that with more production (supply), prices decrease to a certain degree (even on the world market depending on the size of the regions where intensification takes place) – hence there are fewer incentives for using more cropland compared to e.g. pasture land. Note, that the intensification and the reference scenario do not assume cropland expansion into unmanaged areas. Some CGE studies (Château, Dellink et al. 2014) use a land supply curve to simulate expansion into unmanaged land depending on crop price: the higher the prices of crops are, the more expansion into unmanaged land occurs. While the elasticities are not well elaborated in the literature, we preferred to run an exogenous scenario on expansion.

We deliberately analyze cropland intensification and cropland expansion separately in order to be able to quantify the relative differences in the impact of these two alternative global farming strategies.

(4) For an article designed to identify the impacts of agricultural management on biodiversity, it was disappointing not to see potential policy drivers of change being unpicked. For instance, (i) intensification coupled with land planning/zoning that prevents conversion and (ii) a role of other markets that fund protection vs farming, amongst others.

Response: Even though it was not our original intension to incorporate different policies in our scenarios, we think it is worth extending our study as you suggested. We already describe in the Supplement the overlap of expansion and intensification areas with IUCN protected areas for different categories of protection (strict and less strict). The results showed large overlap with expansion (18%) and only small overlap with intensification

areas (4%). Therefore, we now additionally created a policy scenario for expansion that restricts any expansion of cropland in protected areas. The results of this new analysis are now briefly included in the fourth paragraph of the discussion of the main paper and in the Supplementary Information (S7).

References

- Baldos, U. L. C. and T. W. Hertel (2014). "Global food security in 2050: the role of agricultural productivity and climate change." *Australian Journal of Agricultural and Resource Economics* 58(4): 554-570.
- Beckmann, M., K. Gerstner, M. Akin-Fajiyee, S. Ceaușu, S. Kambach, N. Kinlock, H. Phillips, W. Verhagen, J. Gurevitch, S. Klotz, T. Newbold, P. Verburg, M. Winter and R. Seppelt (2019). "Conventional land-use intensification reduces species richness and increases production: A global meta-analysis." *Global Change Biology*. In Press.
- Calzadilla, A., R. Delzeit and G. Klepper (2016). Assessing the Effects of Biofuel Quotas on Agricultural Markets. *The WSPC Reference on Natural Resources and Environmental Policy in the Era of Global Change*. A. Calzadilla, R. Delzeit and G. Klepper, World Scientific: 399-442.
- Château, J., R. Dellink and E. Lanzi (2014). An Overview of the OECD ENV-Linkages Model. *OECD Environment Working Papers No. 65*.
- De Frutos, A., P. P. Olea and P. Mateo-Tomás (2015). "Responses of medium- and large-sized bird diversity to irrigation in dry cereal agroecosystems across spatial scales." *Agriculture, Ecosystems & Environment* 207: 141-152.
- Delzeit, R., F. Zabel, C. Meyer and T. Václavík (2017). "Addressing future trade-offs between biodiversity and cropland expansion to improve food security." *Regional Environmental Change* 17(5): 1429-1441.
- Egli, L., C. Meyer, C. Scherber, H. Kreft and T. Tschardtke (2018). "Winners and losers of national and global efforts to reconcile agricultural intensification and biodiversity conservation." *Global Change Biology* 24(5): 2212-2228.
- Geiger, F., J. Bengtsson, F. Berendse, W. W. Weisser, M. Emmerson, M. B. Morales, P. Ceryngier, J. Liira, T. Tschardtke, C. Winqvist, S. Eggers, R. Bommarco, T. Pärt, V. Bretagnolle, M. Plantegenest, L. W. Clement, C. Dennis, C. Palmer, J. J. Oñate, I. Guerrero, V. Hawro, T. Aavik, C. Thies, A. Flohre, S. Hänke, C. Fischer, P. W. Goedhart and P. Inchausti (2010). "Persistent negative effects of pesticides on biodiversity and biological control potential on European farmland." *Basic and Applied Ecology* 11(2): 97-105.
- Kehoe, L., A. Romero-Muñoz, E. Polaina, L. Estes, H. Kreft and T. Kuemmerle (2017). "Biodiversity at risk under future cropland expansion and intensification." *Nature Ecology & Evolution* 1(8): 1129-1135.
- Kier, G., H. Kreft, T. M. Lee, W. Jetz, P. L. Ibisch, C. Nowicki, J. Mutke and W. Barthlott (2009). "A global assessment of endemism and species richness across island and mainland regions." *Proceedings of the National Academy of Sciences of the United States of America* 106(23): 9322-9327.
- Meehan, T. D., B. P. Werling, D. A. Landis and C. Gratton (2011). "Agricultural landscape simplification and insecticide use in the Midwestern United States." *Proceedings of the National Academy of Sciences* 108(28): 11500-11505.

- Monfreda, C., N. Ramankutty and J. A. Foley (2008). "Farming the planet: 2. Geographic distribution of crop areas, yields, physiological types, and net primary production in the year 2000." Global Biogeochemical Cycles **22**(1): GB1022.
- Müller, C., J. Elliott, J. Chryssanthacopoulos, A. Arneth, J. Balkovic, P. Ciais, D. Deryng, C. Folberth, M. Glotter, S. Hoek, T. Iizumi, R. C. Izaurralde, C. Jones, N. Khabarov, P. Lawrence, W. Liu, S. Olin, T. A. M. Pugh, D. Ray, A. Reddy, C. Rosenzweig, A. C. Ruane, G. Sakurai, E. Schmid, R. Skalsky, C. X. Song, X. Wang, A. de Wit and H. Yang (2017). "Global Gridded Crop Model evaluation: benchmarking, skills, deficiencies and implications." Geoscientific Model Development **10**(4): 1403-1422.
- Ramankutty, N., A. T. Evan, C. Monfreda and J. A. Foley (2008). "Farming the planet: 1. Geographic distribution of global agricultural lands in the year 2000." Global Biogeochemical Cycles **22**(1): GB1003.
- Settele, J., K. L. Heong, I. Kühn, S. Klotz, J. H. Spangenberg, G. Arida, A. Beurepaire, S. Beck, E. Bergmeier, B. Burkhard, R. Brandl, J. V. Bustamante, A. Butler, J. Cabbigat, X. C. Le, J. L. A. Catindig, V. C. Ho, Q. C. Le, K. B. Dang, M. Escalada, C. Dominik, M. Franzén, O. Fried, C. Görg, V. Grescho, S. Grossmann, G. M. Gurr, B. A. R. Hadi, H. H. Le, A. Harpke, A. L. Hass, N. Hirneisen, F. G. Horgan, S. Hotes, Y. Isoda, R. Jahn, H. Kettle, A. Klotzbücher, T. Klotzbücher, F. Langerwisch, W.-H. Loke, Y.-P. Lin, Z. Lu, K.-Y. Lum, D. B. Magcale-Macandog, G. Marion, L. Marquez, F. Müller, H. M. Nguyen, Q. A. Nguyen, V. S. Nguyen, J. Ott, L. Penev, H. T. Pham, N. Radermacher, B. Rodriguez-Labajos, C. Sann, C. Sattler, M. Schädler, S. Scheu, A. Schmidt, J. Schrader, O. Schweiger, R. Seppelt, K. Soitong, P. Stoev, S. Stoll-Kleemann, V. Tekken, K. Thonicke, B. Tilliger, K. Tobias, Y. Andi Trisyono, T. T. Dao, T. Tschardtke, Q. T. Le, M. Türke, T. Václavík, D. Vetterlein, S. B. Villareal, K. C. Vu, Q. Vu, W. W. Weisser, C. Westphal, Z. Zhu, M. J. P. Wiemers and W. Environment (2018). "Rice ecosystem services in South-east Asia." Paddy and Water Environment **16**(2): 211-224.
- Siebert, S., M. Kummu, M. Porkka, P. Döll, N. Ramankutty and B. R. Scanlon (2015). "A global data set of the extent of irrigated land from 1900 to 2005." Hydrology and Earth System Sciences **19**(3): 1521-1545.
- Spangenberg, J. H., A. L. Beurepaire, E. Bergmeier, B. Burkhard, H. Van Chien, L. Q. Cuong, C. Görg, V. Grescho, L. H. Hai, K. L. Heong, F. G. Horgan, S. Hotes, A. Klotzbücher, T. Klotzbücher, I. Kühn, F. Langerwisch, G. Marion, R. F. A. Moritz, Q. A. Nguyen, J. Ott, C. Sann, C. Sattler, M. Schädler, A. Schmidt, V. Tekken, T. D. Thanh, K. Thonicke, M. Türke, T. Václavík, D. Vetterlein, C. Westphal, M. Wiemers and J. Settele (2018). "The LEGATO cross-disciplinary integrated ecosystem service research framework: an example of integrating research results from the analysis of global change impacts and the social, cultural and economic system dynamics of irrigated rice production." Paddy and Water Environment **16**(2): 287-319.
- Vaclavik, T., F. Langerwisch, M. Cotter, J. Fick, I. Häuser, S. Hotes, J. Kamp, J. Settele, J. H. Spangenberg and R. Seppelt (2016). "Investigating potential transferability of place-based research in land system science." Environmental research letters **11**:16.
- Zabel, F., T. Hank and W. Mauser (2019). AgMIP's global gridded crop model intercomparison (GGCMI) phase II CTWN-A archive: priority 1 outputs from PROMET maize simulations. Zenodo. doi:10.5281/zenodo.2582467.

REVIEWERS' COMMENTS:

Reviewer #1 (Remarks to the Author):

Dear authors,
thank you for improving your paper and your detailed answers.

Line 376: please write "addressed" instead of "addresses"

Fig. 2: If green and orange are printed in black and white, both look the same. That's why other colours would have been better! But maybe it's not important.

You have addressed all of the comments. In my opinion, the manuscript was greatly improved.

Reviewer #2 (Remarks to the Author):

The authors have done a fine job responding to all three reviewers' concerns, in my opinion, and the paper makes a useful contribution to the literature.

Reviewer #3 (Remarks to the Author):

I am happy with the authors revisions

Response to reviewer remarks

REVIEWERS' COMMENTS:

Reviewer #1 (Remarks to the Author):

Dear authors,
thank you for improving your paper and your detailed answers.

Line 376: please write "addressed" instead of "addresses"

Answer: many thanks, it is changed accordingly.

Fig. 2: If green and orange are printed in black and white, both look the same. That's why other colours would have been better! But maybe it's not important.

Answer: We would prefer to keep it as it is to match the used colors in the other maps, tables and figures. If the editor would like us to change it, please let us know.

You have addressed all of the comments. In my opinion, the manuscript was greatly improved.

Reviewer #2 (Remarks to the Author):

The authors have done a fine job responding to all three reviewers' concerns, in my opinion, and the paper makes a useful contribution to the literature.

Reviewer #3 (Remarks to the Author):

I am happy with the authors revisions